# Phenotyping for Effects of Drought Levels in Quinoa Using Remote Sensing Tools

Nerio E. Lupa-Condo [1], Frans C. Lope-Ccasa [1], Angel A. Salazar-Joyo [1], Raymundo O. Gutiérrez-Rosales [2], Eric N. Jellen [3], Neil C. Hansen [3], Alberto Anculle-Arenas [1], Omar Zeballos [1], Natty Wilma Llasaca-Calizaya [4] and Mayela Elizabeth Mayta-Anco [1,*]

1   Facultad de Agronomía, Universidad Nacional de San Agustín de Arequipa, Arequipa 04001, Peru; nlupa@unsa.edu.pe (N.E.L.-C.); flopec@unsa.edu.pe (F.C.L.-C.); asalazarj@unsa.edu.pe (A.A.S.-J.); lanculle@unsa.edu.pe (A.A.-A.); ozeballos@unsa.edu.pe (O.Z.)
2   Facultad de Ingeniería Agrícola, Universidad Nacional Agraria de La Molina, Lima 15012, Peru; 19910125@lamolina.edu.pe
3   Department of Plant and Wildlife Sciences, College of Life Sciences, Brigham Young University, Provo, UT 84602, USA; rick_jellen@byu.edu (E.N.J.); neil_hansen@byu.edu (N.C.H.)
4   Facultad de Ciencias Histórico Sociales, Universidad Nacional de San Agustín de Arequipa, Arequipa 04001, Peru; nllasaca@unsa.edu.pe
*   Correspondence: mmaytaan@unsa.edu.pe

**Abstract:** Drought is a principal limiting factor in the production of agricultural crops; however, quinoa possesses certain adaptive and tolerance factors that make it a potentially valuable crop under drought-stress conditions. Within this context, the objective of the present study was to evaluate morphological and physiological changes in ten quinoa genotypes under three irrigation treatments: normal irrigation, drought-stress followed by recovery irrigation, and terminal drought stress. The experiments were conducted at the UNSA Experimental Farm in Majes, Arequipa, Peru. A series of morphological, physiological, and remote measurements were taken, including plant height, dry biomass, leaf area, stomatal density, relative water content, selection indices, chlorophyll content via SPAD, multispectral imaging, and reflectance measurements via spectroradiometry. The results indicated that there were numerous changes under the conditions of terminal drought stress; the yield variables of total dry biomass, leaf area, and plant height were reduced by 69.86%, 62.69%, and 27.16%, respectively; however, under drought stress with recovery irrigation, these changes were less pronounced with a reduction of 21.10%, 27.43%, and 17.87%, respectively, indicating that some genotypes are adapted or tolerant of both water-limiting conditions (Accession 50, Salcedo INIA and Accession 49). Remote sensing tools such as drones and spectroradiometry generated reliable, rapid, and precise data for monitoring stress and phenotyping quinoa and the optimum timing for collecting these data and predicting yield impacts was from 79–89 days after sowing (NDRE and CREDG r Pearson 0.85).

**Keywords:** quinoa; drought; multispectral imaging; reflectance index; vegetation index

## 1. Introduction

Worldwide agriculture is facing constant challenges from climate change, which has increased the frequency of extreme temperature, salinity, hail, and drought episodes [1,2]. These phenomena pose greater productivity risks to certain crops and varieties, thus threatening food security and agricultural sustainability [3–7]. In particular, drought events have become more frequent, intense, and prolonged due to climate change [8–10], and thus represent a significant environmental challenge [11]. Specifically, agricultural drought, which is defined as a prolonged period of water deficit in the soil [12], is one of the main abiotic stress factors adversely affecting crop production in distinct areas worldwide [13], diminishing the growth and development of crop plants [14,15]. In this context, quinoa

(*Chenopodium quinoa* Willd.), an ancient pseudocereal domesticated in the Andean countries for the past 7000 years [16], stands out as a promising option for sustainable food production and ensuring food security [17,18] due to its superior nutritional value and ability to tolerate adverse environmental conditions, including drought [19].

Studies on the effects of drought reveal that this type of stress negatively alters diverse aspects of the plant, among them: photosynthesis through the disruption of chloroplasts, including the formation of thylakoid membranes and the function of the plastid electron transport chain [15]; plant growth; grain yield; relative water content [20–22]; root growth and development [23]; biomass; stomatal density; and leaf hydration [24]. However, some crops possess distinct adaptive mechanisms for tolerating or resisting abiotic stresses through physiological, morphological, and phenological strategies [7,9], among which is quinoa, which is considered to be highly resistant to multiple abiotic stresses [25]. Nevertheless, there exists a wide range of variation for drought tolerance among quinoa genotypes [9], many of which are grown by small-scale subsistence farmers and have high levels of biodiversity [26], which explains the importance of seeking out and selecting these highly resistant genotypes.

Understanding drought-induced physiological and morphological changes is crucial for genetic improvement and management strategies to improve drought stress tolerance and production for quinoa. The evaluation of the effects of water stress in quinoa has traditionally been based on direct observation and in situ measurements and it is limited by time and space considerations [27]. In contrast, remote sensing tools can offer a promising alternative for monitoring plant changes in a non-invasive way [28]; the rapid adjustment of data and capacity to detect intervarietal differences under water stress conditions highlight the power of remote detection for gathering phenotypic data [29]. However, in plants that do not attain high levels of vegetative density, remote sensing using drones may not be feasible due to interfering influences from the soil [30].

The objective of this study was to evaluate morphological and physiological changes in quinoa genotypes under three irrigation treatments—normal irrigation, drought with irrigation-mediated recovery, and terminal drought—in an arid production zone. The study involved conventional evaluation methods in the field as well as advanced remote sensing and telemetry tools as complementary approaches for monitoring and gathering digital data on quinoa cultivation.

## 2. Materials and Methods

### 2.1. Experimental Site

The agronomic experiment was conducted in the field between July and December, 2023, at the Centro de Investigación, Enseñanza y Producción Agrícola (CIEPA) (16°19′30″ S, 72°13′01″ W, and 1429 m above sea level), at the Universidad Nacional de San Agustín de Arequipa, Arequipa, Perú. During the experimental period, the average temperature was 19.20 °C, with a maximum temperature of 27.7 °C and a minimum of 7.7 °C. The average relative humidity was 42.53% and the accumulated precipitation was 3.1 mm (Servicio Nacional de Meteorología e Hidrología (SENAMHI) de Pampa de Majes, Estación Meteorológica Automática) (Figure 1). The soil at the experimental site was sandy loam with a pH of 7.5, electrical conductivity of 0.938 dS m$^{-1}$, and an organic matter content of 1.1%. The irrigation water had a pH of 8.0 and electrical conductivity of 0.70 dS m$^{-1}$ (Instituto Nacional de Innovación Agraria (INIA)-Water, Soil and Foliar Research Laboratory).

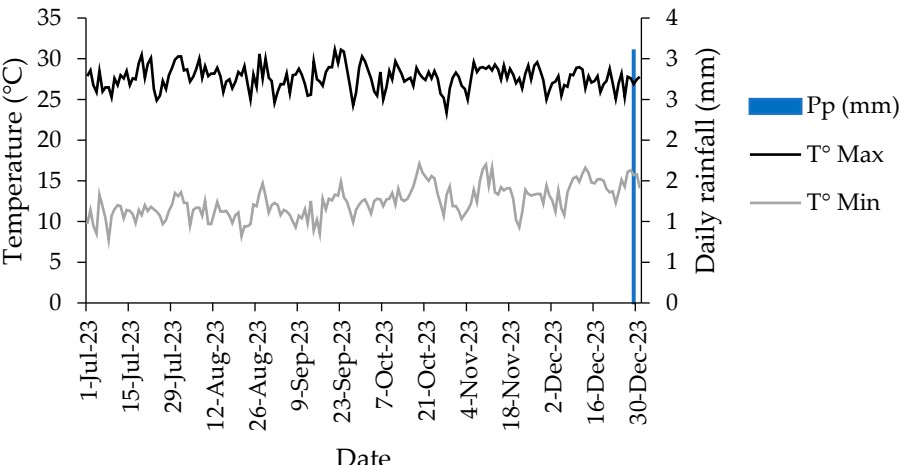

**Figure 1.** Daily rainfall (Pp) and maximum and minimum temperature (T° Max, T° Min) at Estación Meteorológica Automática Pampa de Majes in Majes, Arequipa, Peru.

*2.2. Plant Materials*

The experiment featured 10 genotypes, including 2 commercial varieties released by INIA plus 8 *Chenopodium quinoa* Willd. strains from the Banco de Germoplasma Project of the Universidad Nacional de San Agustín de Arequipa (Table 1).

**Table 1.** Genetic material utilized in this experiment.

| Genotype | GB UNSA Code [1] |
| --- | --- |
| Accession 06 | UNSA-CH-1900006 |
| Accession 20 | UNSA-CH-1900020 |
| Accession 23 | UNSA-CH-1900023 |
| Accession 26 | UNSA-CH-1900026 |
| Accession 42 | UNSA-CH-1900042 |
| Accession 43 | UNSA-CH-1900043 |
| Accession 49 | UNSA-CH-1900049 |
| Accession 50 | UNSA-CH-1900050 |
| Blanca de Juli | Blanca de Juli |
| Salcedo INIA | Salcedo INIA |

[1] GB UNSA CODE: Code provided by the Banco de Germoplasma Project of the Universidad Nacional de San Agustín de Arequipa.

*2.3. Experimental Design and Management*

The experiment was conducted as a randomized complete block design (RCBD), arranged in main plots by irrigation treatments: normal irrigation (NI), drought with recovery irrigation (DR), and terminal drought (TD). In the sub-plots were the 10 genotypes, with each treatment including 4 replicates, thus forming a total of 120 experimental units (Figure 2). The area allocated to each experimental unit was 10.8 m². Planting occurred on 30 July 2023; in each experimental unit they were planted in four rows with a spacing of 0.9 m between rows and 0.2 m between strokes (stroke = set of 10 plants), resulting in a density of 50 plants per linear meter. Fertilizers were applied daily via irrigation from 15 days after sowing (DAS) until 63 DAS, using a base fertilizer dose per hectare of 208-110-241-33-27 of N-P-K-Ca-Mg. All three irrigation treatments (NI, DR and TD) were fertilized at the same rates.

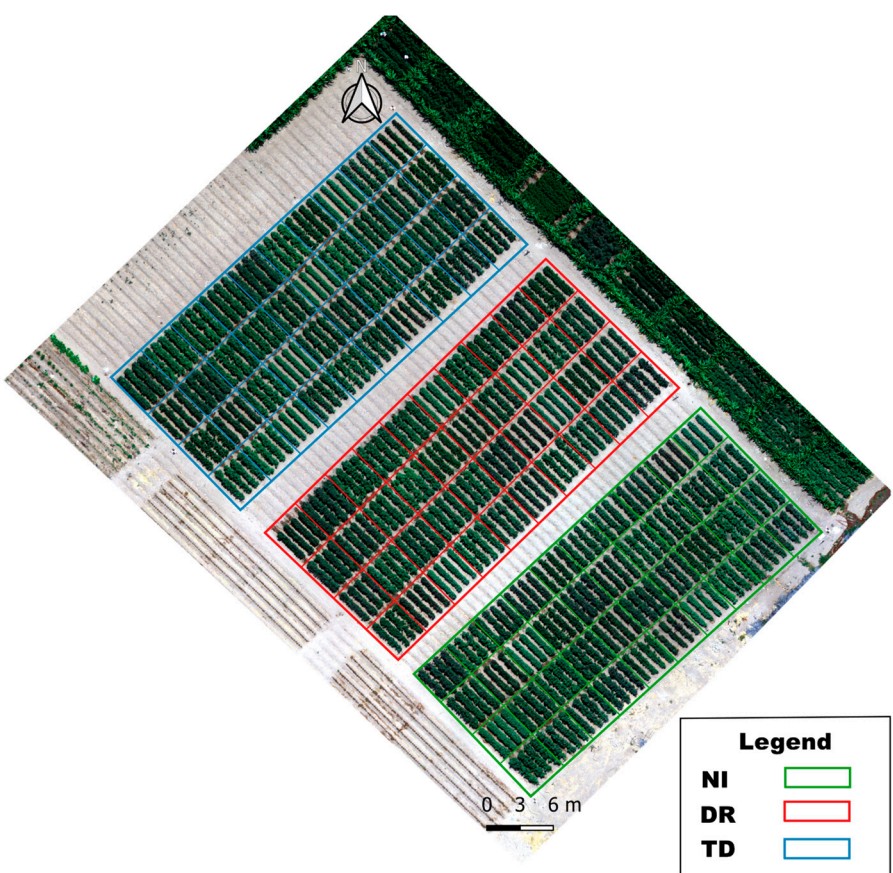

**Figure 2.** UAV image of the experimental field and plots 63 days after sowing. Abbreviations: NI = normal irrigation; DR = drought with recovery irrigation; and TD = terminal drought.

Irrigation was applied daily in the three plots until 63 DAS. With NI, water was applied daily up until the point of harvest; for DR, irrigation was suspended at 64 DAS, resumed 15 days after drought (79 DAS), and continued up until the harvest; and for TD, watering was suspended at 64 DAS and was not resumed prior to harvest (Figure 3). Irrigation water was applied in quantities of 409.25, 339.71, and 102.5 m$^3$ for NI, DR, and TD treatments, respectively. The harvest of the quinoa genotypes was carried out according to the physiological maturity reached; Accession 50 and Accession 49 were harvested 100 DAS, and the other genotypes were harvested at 136 DAS.

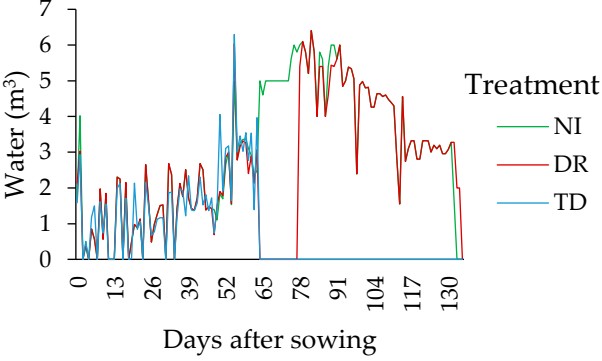

**Figure 3.** Daily application of water for the three irrigation treatments. Abbreviations: NI = normal irrigation; DR = drought with recovery irrigation; and TD = terminal drought.

### 2.4. Morphological Evaluations

One of the effects of drought on plants is on their morphological aspects; within this, leaves are one of the direct indicators of water scarcity [31]. For this reason, within the morphological variables, plant height and leaf area were evaluated. Height was measured on 10 randomly selected plants from the central rows of each experimental unit. Using a tape measure, the distance from the soil surface to the plant apex was measured. Panicle (inflorescence) length (PL) and diameter (PD) were measured at physiological maturity in the same plants selected for height measurements.

Leaf area (cm$^2$ per plant) was evaluated at 77 DAS (14 d after drought stress) and 88 DAS (25 d after drought stress for TD and 11 d after recovery irrigation for DR treatments). Plants from the central rows were selected from each experimental unit; later, the leaves were separated from the remainder of the plant and scanned on a flat scanner. Foliar area (LA) was calculated using Compu Eye, Leaf & Symptom Area software version 1.0 (Doki, Cairo, Egypt) in cm$^2$ [32].

### 2.5. Physiological Evaluations

The leaf relative water content (LRWC) and specific leaf area (SLA) were measured at 73 DAS (10 days post-stress for DR and TD treatments) and 94 DAS (31 days post-stress for TD and 17 days after irrigation recovery for DR). Three leaves were collected from different plants of each experimental treatment. Fresh weight (FW) of the leaves was measured immediately. Afterwards, the leaves were submerged in distilled water in a plastic tray for 24 h to determine turgid weight (TW). Lastly, the leaves were scanned to calculate specific leaf area, and dry weight (DW) was determined after drying them at 80 °C for 48 h. Specific leaf area was calculated as the coefficient between the leaf area (cm$^2$) and the dry weight value (g), while leaf relative water content (LRWC) was calculated using the formula of Barrs & Weatherley [33] as follows:

$$\text{LRWC}(\%) = (\text{FW} - \text{DW}) / (\text{TW} - \text{DW}) \times 100 \tag{1}$$

Chlorophyll content was measured using a SPAD 502 Plus (Konica Minolta, Osaka, Japón) chlorophyllometer instrument by the method of Hurtado et al. [34] with several modifications. Readings were taken on fully extended and healthy leaves from the upper third of the plants. In total, measurements were taken on 10 plants per experimental unit, sampling three leaves per plant and taking leaf-area readings from three different zones of each leaf; the average of these measurements were then calculated for each plant. Chlorophyll measurements were taken at 44, 59, 70, 83, and 106 DAS.

To determine stomatal density (SD) and stomatal index (SI), three leaves were chosen from the upper third of different plants within each experimental unit. The leaves were washed with distilled water, air dried, then transparent varnish was applied to the abaxial surfaces. After drying, the layer was removed with transparent adhesive tape and they were mounted on slides for observation with a compound microscope (Amscope, Irvine, CA, USA) at 40× magnification. The SD was determined by counting the number of stomata within an area of $2.596 \times 10^{-5}$ mm$^2$, while the SI was calculated using the following equation proposed by Wilkinson [35]:

$$\text{SI} = (\text{SN} * 100) / (\text{CN} - \text{SN}) \tag{2}$$

where SN and CN represent stomatal and epidermal cell numbers, respectively, within the microscopic field of observation.

### 2.6. Evaluations with Multispectral Remote Sensors

Multispectral images were acquired using a Phantom 4 Multispectral aerial drone (UAV; DJI, Shenzhen, China) having an integrated six-camera system, consisting of an RGB visible camera plus five multispectral cameras covering the following wavelengths: blue (B): $450 \pm 16$ nm; green (G): $560 \pm 16$ nm; red (R): $650 \pm 16$ nm; far-red (RE): $730 \pm 16$ nm;

and near infrared (NIR): 840 ± 26 nm. Flights were conducted at 30 m, with a frontal and lateral overlap of 75%, a camera angle of 90°, and with a spatial resolution of 1.6 cm pixel$^{-1}$. A total of 18 flights were made at differing phenological stages of the crop.

The orthomosaics and the Digital Elevation Model (DEM) were generated using the photogrammetry software Agisoft Metashape version 2.0.3 build 16915. Afterwards, the orthomosaics thus produced were integrated using the free and open-access Geographic Information System (GIS) software QGIS version 2.28.12, where the alignment was carried out using five ground control points (GCP) previously arranged in the experimental field. Next, the area of interest was isolated by extracting the polygonal masked layer. Then, the soil vegetation was segmented using a threshold value of 0.12 for the Optimized Soil Adjusted Vegetation Index (OSAVI). Grids were created for the two central rows of each experimental unit and finally, the average pixel values were recorded for each grid. Based on these values, except for Photochemical Reflectance Index (PRI), 24 vegetation indices were calculated (Table 2) along with the percent of canopy coverage (CC). Figure 4 shows the image of NDVI calculated at 63, 77, and 86 DAS (before, during and after drought).

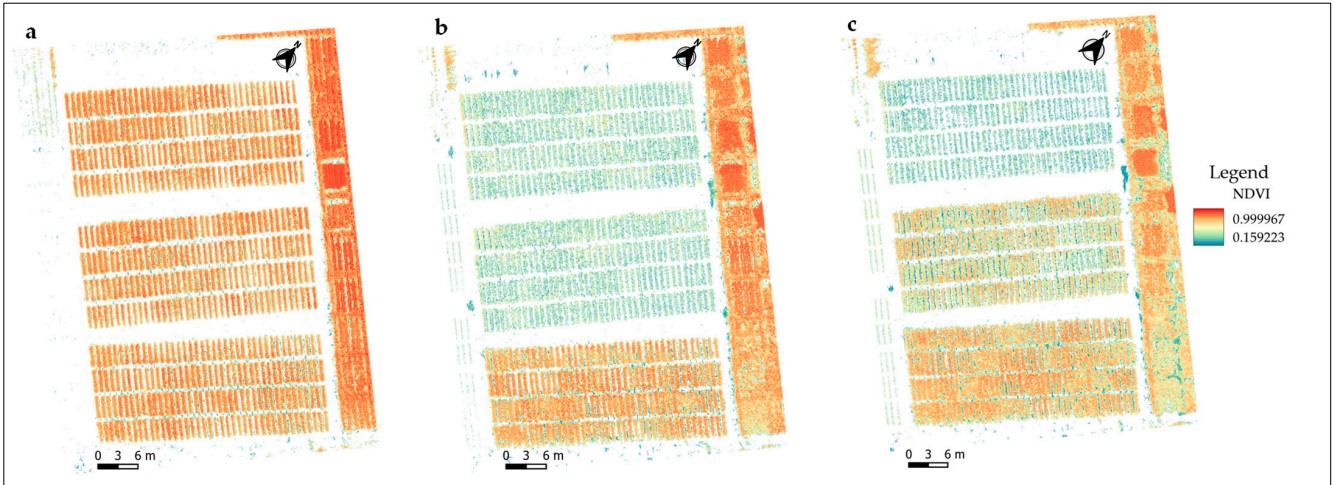

**Figure 4.** Normalized Difference Vegetation Index (NDVI) image of the plots under three irrigation treatments: (**a**) NDVI image at 63 DAS (1 day before the start of drought treatment); (**b**) NDVI image at 77 DAS (14 days after drought); (**c**) NDVI image at 86 DAS (23 days after drought for TD and 9 days after recovery irrigation for DR).

Spectroradiometric measurements were performed following the method of Chávez et al. [36] with modifications. An AvaSpec spectrometer (ULS2048CL-EVO, Avantes, Apeldoorn, The Netherlands) having a measurement range from 441 to 984 nm and bandwidth (spectral resolution) of ±0.29 nm was mounted on a tripod and connected to a laptop computer; the calibration and readings were performed using AvaSoft version 8.15.0.0 software. Measurements were collected from 10:00 to 11:00 and from 14:00 to 15:00. The measurement distance between the nadir and plant canopy was 30 cm, which resulted in a circular field of vision of 12.12 cm for the optical angle of 22°; four measurements per experimental unit were taken. The measurements were taken at 73, 86, and 100 DAS. Sixteen vegetation indices were taken using the formulas found in the library of formulas in Index DataBase. Available online: https://www.indexdatabase.de (accessed on 23 January 2024), which are also reported in Table 2.

**Table 2.** Vegetation indices calculated during the experiments.

| ID | Index | Description | Formula | Reference |
|----|-------|-------------|---------|-----------|
| 1 | NDVI | Normalized Difference Vegetation Index | $\frac{NIR-RED}{NIR+RED}$ | [37] |

**Table 2.** *Cont.*

| ID | Index | Description | Formula | Reference |
|---|---|---|---|---|
| 2 | TVI | Transformed Vegetation Index | $\left(\frac{(NIR-RED)}{(NIR+RED)} + 0.5\right)^{0.5}$ | [38] |
| 3 | RVI | Ratio Vegetation Index | $\frac{NIR}{Red}$ | [39] |
| 4 | MRVI | Modified Normalized Difference Vegetation Index | $\left(\frac{NIR}{Red} - 1\right) / \left(\frac{NIR}{Red} + 1\right)$ | [40] |
| 5 | SAVI_CC | Soil Adjusted Vegetation Index | $\left(\frac{NIR-RED}{NIR+RED+CC}\right) \times (1+CC)$ | [41] |
| 6 | SAVI_05 | Soil and Atmospherically Resistant Vegetation Index | $\left(\frac{NIR-RED}{NIR+RED+0.5}\right) \times (1+0.5)$ | [42] |
| 7 | AVI | Ashburn Vegetation Index | $NIR - RED$ | [43] |
| 8 | IPVI | Infrared Percentage Vegetation Index | $\frac{NIR}{NIR+RED}$ | [44] |
| 9 | CTVI | Corrected Transformed Vegetation Index | $\left(\frac{NDVI+0.5}{ABS(NDVI+0.5)}\right) \times (ABS(NDVI+0.5))^{0.5}$ | [45] |
| 10 | GNDVI | Green Normalized Difference Vegetation Index | $\left(\frac{NIR-GREEN}{NIR+GREEN}\right)$ | [46] |
| 11 | LCI | Leaf Chlorophyll Index | $\left(\frac{NIR-REDEGDE}{(NIR+RED}\right)$ | [47] |
| 12 | CIG | Chlorophyll Index Green | $\left(\frac{NIR}{GREEN}\right) - 1$ | [48] |
| 13 | CIREDG | Chlorophyll Index RedEdge | $\left(\frac{NIR}{REDEDGE}\right) - 1$ | [48] |
| 14 | CCCI | Canopy Chlorophyll Content Index | $\left(\frac{NIR-REDEDGE}{NIR+REDEDGE}\right) / \left(\frac{NIR-RED}{NIR+RED}\right)$ | [49] |
| 15 | NDRE | Normalized Difference Red Edge index | $\left(\frac{NIR-REDEDGE}{NIR+REDEDGE}\right)$ | [49] |
| 16 | CREDG | Chlorophyll Red-Edge | $\left(\frac{NIR}{REDEDGE}\right)^{-1}$ | [50] |
| 17 | CVI | Chlorophyll Vegetation index | $NIR \times \left(\frac{RED}{GREEN^2}\right)$ | [51] |
| 18 | GRVI | Green Ratio Vegetation Index | $\frac{NIR}{GREEN}$ | [52] |
| 19 | NGRDI | Normalized Green-Red Difference index | $\frac{GREEN-RED}{GREEN+RED}$ | [52] |
| 20 | RGR | Simple Ratio Red/Green | $\frac{RED}{GREEN}$ | [53] |
| 21 | GRE | Green Red Edge index | $GREEN \times REDEDGE$ | [54] |
| 22 | GBNDVI | Green-Blue NDVI | $\left(\frac{NIR-(GREEN+BLUE)}{NIR+(GREEN+BLUE)}\right)$ | [55] |
| 23 | GRNDVI | Green-Red NDVI | $\left(\frac{NIR-(GREEN+RED)}{NIR+(GREEN+RED)}\right)$ | [55] |
| 24 | HI | Health Index | $\left(\frac{GREEN-REDEDGE}{GREEN+REDEDGE}\right) - (0.5 \times REDEDGE)$ | [56] |
| 25 | PRI | Photochemical Reflectance Index | $\frac{RED_{531}-RED_{570}}{RED_{531}+RED_{570}}$ | [57] |

### 2.7. Agronomic Evaluations

Grain yield (t ha$^{-1}$) was calculated from the weight of harvested seed from each experimental unit. The seed thickness (TS) and seed diameter (DS) were measured in millimeters (mm); to calculate TS a micrometer was used and for DS, a NexiusZoom stereomicroscope (Euromex, Arnhem, The Netherlands). Thousand seed weight (TSW) was determined using an analytical balance model AS R2 PLUS (Radwag, Radom, Poland,

±0.0001 g) and was expressed in grams. Harvest index (HI) was calculated as the ratio of grain yield (kg) to total plant biomass (kg).

In determining dry biomass (BM), three plants from each experimental unit were removed and separated as roots, stems, and inflorescences (panicles), which were collected in paper bags. To measure dry weight, the samples were placed in an 80 °C oven for 48 h. Total biomass (BT, t ha$^{-1}$) was the sum of the dry weight of the separated parts of the plant. Water use efficiency (WUE, kg m$^{-3}$) for each treatment was calculated as the ratio between the total biomass yield (kg ha$^{-1}$) and the water applied per irrigation treatment (m$^3$ ha$^{-1}$).

### 2.8. Evaluations of Selection Indices

The Drought Tolerance Index (DTI) and Drought Susceptibility Index (DSI) were calculated for the DR and TD irrigation treatments, accounting for NI. The DTI was determined utilizing the formula of Fernández [58]; while the DSI was calculated following the formula of Fischer & Maurer [59]:

$$DTI = (Y_p \times Y_s)/(\overline{Y}_p)2 \tag{3}$$

$$SI = (1 - (Y_s/Y_p))/(1 - (\overline{Y}_s/\overline{Y}_p)) \tag{4}$$

where $Y_s$ and $Y_p$ represent the genotype yields with and without stress, while the values $\overline{Y}_s$ y $\overline{Y}_p$ represent the average yield of the genotypes under stress and non-stress conditions, respectively.

### 2.9. Statistical Analyses

R software version 4.3.3 [60] and RStudio version 2023.09.1+494 were used to process the data. The data for each variable were subjected to ANOVA, evaluating the differences among the means of each genotype under each irrigation treatment and the differences between irrigation treatments was assessed using the Duncan's comparative means test (0.05) in the R software package, Agricolae [61]. Corrplot [62] was used to determine the correlation matrix and the R software packages, factoMineR [63] and factoextra [64] were used for conducting the principal component analysis (PCA).

## 3. Results

### 3.1. Effect of Drought on Morphological Variables

Plant growth for the different irrigation treatments did not differ significantly until 67 DAS. However, after this point plants were taller under the NI treatment than under DR and TD; these last two significant differences were observed beginning at 87 DAS (Figure 5a). In terms of leaf area, at 76 DAS in NI, it was significantly greater than in the DR and TD treatments, while these stress treatments showed no significant differences. However, at 88 DAS there were significant differences in leaf area for all three irrigation treatments (Figure 5b).

Table 3 contains data on morphological variables. For plant height, Accession 43 had the greatest values across all three treatments. Accession 50 was shortest under the NI treatment, while the cultivar Blanca de Juli was shortest in the DR and TD treatments. The different irrigation treatments affected plant height in all genotypes except for Accession 50, which maintained similar heights under all three treatments. For leaf area, Accession 43 had the largest values in all three treatments, while the smallest leaf area values were shared by Blanca de Juli and Accession 50. The cultivar Blanca de Juli showed the least amount of variation for leaf area under the three irrigation regimes.

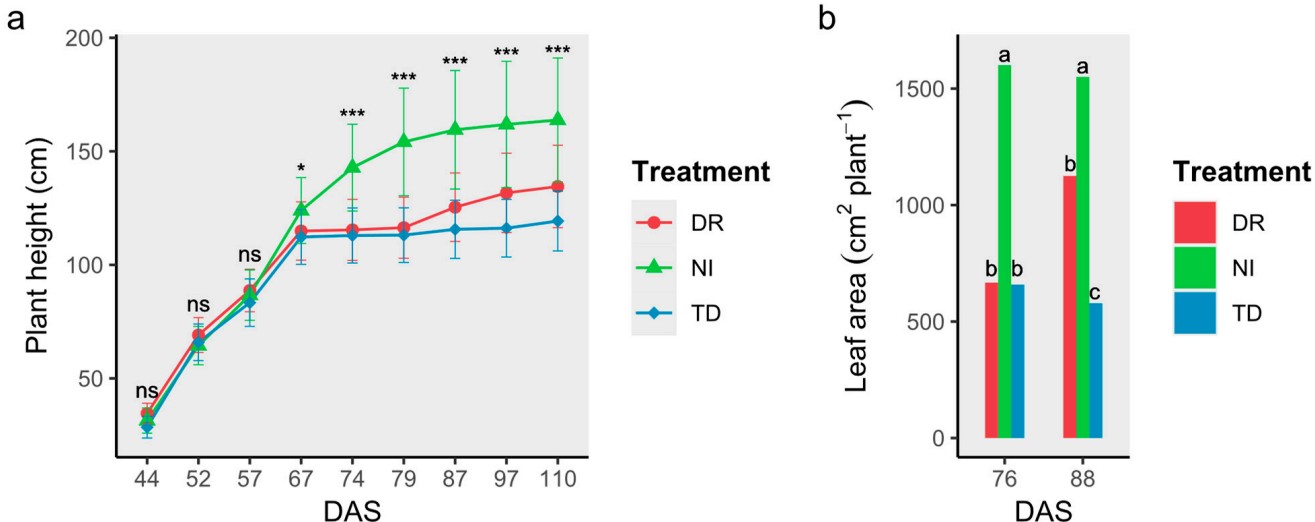

**Figure 5.** Morphological variables: (**a**) progression of plant height under the three irrigation treatments; (**b**) leaf area for the irrigation treatments at 76 and 88 DAS. The asterisks above the symbols indicate significant treatment differences: *, *** F test significant at $p \leq 0.05$ and 0.001, respectively; ns = not significant. Letters above the columns indicate significant differences among irrigation treatments ($p \leq 0.05$, Duncan's test).

**Table 3.** Morphological variables of the 10 genotypes in the three irrigation treatments.

| Genotypes | Plant Height (cm) | | | Leaf Area (cm² Plant⁻¹) | | | PL (cm) | | | PD (cm) | | |
|---|---|---|---|---|---|---|---|---|---|---|---|---|
| | NI | DR | TD | NI | DR | DT | NI | DR | TD | NI | DR | TD |
| Accession 06 | 165.80 cd | 133.08 b | 106.43 d | 1861.94 ab | 709.61 c | 522.90 bc | 68.25 a | 51.79 ab | 21.42 b | 9.75 ab | 10.13 ab | 4.10 a |
| Accession 20 | 171.63 bc | 135.55 b | 115.45 cd | 1518.47 bc | 1368.88 b | 534.37 bc | 69.67 a | 59.50 a | 25.08 b | 10.85 a | 11.79 a | 4.63 a |
| Accession 23 | 162.55 cd | 134.83 b | 125.00 abc | 1240.32 bc | 1319.16 b | 514.38 bc | 66.75 a | 45.67 bc | 20.17 b | 11.38 a | 11.04 ab | 4.18 a |
| Accession 26 | 171.20 bc | 133.33 b | 116.25 bcd | 1326.62 bc | 1408.02 b | 544.89 bc | 52.04 bc | 44.71 bc | 21.58 b | 10.42 a | 10.33 ab | 4.64 a |
| Accession 42 | 184.05 b | 135.65 b | 122.75 abc | 2390.84 a | 947.13 bc | 648.59 bc | 52.96 bc | 41.83 c | 23.29 b | 7.92 bc | 8.17 bc | 4.35 a |
| Accession 43 | 212.53 a | 164.03 a | 133.15 a | 2392.29 a | 2060.85 a | 1200.26 a | 69.58 a | 57.21 a | 21.92 b | 10.92 a | 11.63 a | 4.94 a |
| Accession 49 | 150.58 d | 132.93 b | 127.95 ab | 1216.90 bc | 1014.09 bc | 390.91 bc | 56.04 b | 43.96 bc | 34.25 a | 8.92 ab | 8.83 ab | 4.16 a |
| Accession 50 | 121.55 e | 122.75 bc | 124.10 abc | 1194.28 bc | 651.44 c | 367.78 bc | 46.17 c | 42.42 c | 34.63 a | 6.13 c | 5.96 c | 3.66 a |
| Blanca de Juli | 133.90 e | 108.38 c | 95.60 e | 877.69 c | 731.02 c | 299.67 c | 51.71 bc | 42.13 c | 17.63 b | 11.21 a | 11.46 a | 3.78 a |
| Salcedo INIA | 164.23 cd | 144.78 ab | 126.35 abc | 1485.79 bc | 1042.53 bc | 760.80 b | 55.58 b | 46.42 bc | 25.38 b | 9.25 ab | 9.17 ab | 3.46 a |
| Mean | 163.80 | 134.53 | 119.30 | 1550.51 | 1125.27 | 578.45 | 58.88 | 47.56 | 24.53 | 9.67 | 9.85 | 4.19 |

In each column, lowercase letters indicate significant differences among genotypes ($p \leq 0.05$, Duncan's test). Abbreviations: PL = Panicle length; PD = Panicle diameter.

The average value of PL for the genotypes in the NI and DR treatments were similar at 58.88 and 47.56 cm, respectively, while for the TD treatment (24.53 cm) the average value was significantly less when compared to treatments NI and DR. Accessions 20, 43, 06, and 23 had significantly higher values for PL than the other genotypes in the NI treatment while the same group, minus genotype 23, was also significantly higher in treatment DR. While in the TD treatment, the highest values were observed for Accessions 50 and 49, and the lowest value was measured in Blanca de Juli. For the variable PD, values were similar for the NI and DR treatments, except for in Accessions 50 and 42; while in treatment TD, differences among genotypes were not significant. The smallest reduction was observed in Accession 50 with 40.29% in the TD treatment when compared with treatment NI.

### 3.2. Effect of Drought on Physiological Variables

Significant differences were observed for chlorophyll content via SPAD beginning at 59 DAS for the irrigation treatments, with the highest values seen for TD and DR (Figure 6a). With respect to leaf relative water content (LRWC), the highest values were observed for NI; at 73 DAS there were no significant differences between treatments DR

and TD. Nonetheless, at 94 DAS, significant differences were observed among treatments (Figure 6b).

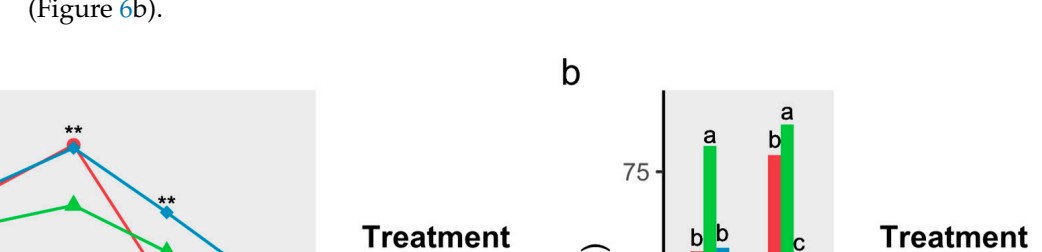

**Figure 6.** Physiological variables: (**a**) chlorophyll content (SPAD) by irrigation treatment; (**b**) leaf relative water content (LRWC) by irrigation treatment at 76 and 88 DAS. Asterisks above the data points indicate significant differences: *, **, *** F test significant at $p \leq 0.05$, 0.01 and 0.001, respectively; ns not significant. Letters above columns in (**b**) denote significant differences among irrigation treatments ($p \leq 0.05$, Duncan's test).

Table 4 presents physiological variables for tested genotypes under the three irrigation treatments. Highest chlorophyll contents were measured in the following genotypes: Accessions 06, 20, 42, 43, 23, and Blanca de Juli under normal irrigation (NI), with Accessions 43 and 42 maintaining superiority under the three irrigation treatments. Accession 50 had the lowest chlorophyll values in the three treatments. As a general rule, strains increased in chlorophyll content under drought stress conditions. Meanwhile, specific leaf area (SLA) values did not differ under irrigation treatments.

**Table 4.** Physiological variables for the 10 genotypes under three irrigation treatments.

| Genotypes | Chlorophyll Content (SPAD) | | | Leaf Relative Water Content (LRWC, %) | | | Specific Leaf Area (SLA, cm² g⁻¹) | | | Stomatal Density (SD) | | |
|---|---|---|---|---|---|---|---|---|---|---|---|---|
| | NI | DR | TD | NI | DR | DT | NI | DR | TD | NI | DR | TD |
| Accession 06 | 47.98 [a] | 49.26 [ab] | 54.03 [bc] | 85.97 [ab] | 80.35 [a] | 63.04 [a] | 181.16 [a] | 189.97 [b] | 189.53 [a] | 33 [ab] | 36 [ab] | 52 [ab] |
| Accession 20 | 45.88 [ab] | 50.19 [ab] | 53.53 [bc] | 83.64 [ab] | 80.84 [a] | 58.43 [a] | 213.14 [a] | 178.20 [b] | 195.01 [a] | 36 [a] | 38 [a] | 58 [ab] |
| Accession 23 | 40.84 [abc] | 46.27 [ab] | 48.68 [cd] | 89.60 [a] | 79.95 [a] | 56.28 [a] | 212.93 [a] | 196.00 [b] | 193.19 [a] | 33 [ab] | 38 [a] | 63 [a] |
| Accession 26 | 36.59 [c] | 46.25 [ab] | 52.75 [bc] | 81.44 [ab] | 77.82 [a] | 48.48 [a] | 254.59 [a] | 209.05 [ab] | 208.21 [a] | 33 [ab] | 31 [b] | 68 [a] |
| Accession 42 | 43.62 [abc] | 47.70 [ab] | 55.70 [ab] | 89.84 [a] | 78.65 [a] | 56.79 [a] | 219.13 [a] | 188.21 [b] | 193.40 [a] | 31 [ab] | 39 [a] | 71 [a] |
| Accession 43 | 42.73 [abc] | 51.75 [a] | 59.92 [a] | 87.07 [ab] | 76.68 [a] | 59.76 [a] | 187.98 [a] | 195.46 [b] | 197.46 [a] | 35 [ab] | 36 [ab] | 79 [a] |
| Accession 49 | 37.48 [c] | 45.03 [b] | 45.75 [d] | 80.82 [ab] | 74.73 [a] | 55.59 [a] | 208.63 [a] | 184.98 [b] | 191.87 [a] | 29 [b] | 32 [b] | 62 [ab] |
| Accession 50 | 36.48 [c] | 39.14 [c] | 33.14 [e] | 75.43 [b] | 72.82 [a] | 61.72 [a] | 239.57 [a] | 248.21 [a] | 228.80 [a] | 19 [c] | 20 [c] | 34 [b] |
| Blanca de Juli | 41.79 [abc] | 51.35 [a] | 50.14 [bcd] | 85.57 [ab] | 80.70 [a] | 50.61 [a] | 198.80 [a] | 192.94 [b] | 182.72 [a] | 33 [ab] | 39 [a] | 56 [ab] |
| Salcedo INIA | 40.32 [bc] | 50.86 [a] | 52.21 [bc] | 91.62 [a] | 83.20 [a] | 64.57 [a] | 228.86 [a] | 189.24 [b] | 186.66 [a] | 33 [ab] | 36 [ab] | 64 [a] |
| Mean | 41.37 | 47.78 | 50.58 | 85.10 | 78.57 | 57.53 | 214.48 | 197.22 | 196.68 | 32 | 34 | 61 |

In each column, lowercase letters indicate significant differences among genotypes ($p \leq 0.05$, Duncan's test).

Average LRWC values under treatments NI and DR were 85.10% and 78.57%, respectively, and declined to 57.53% in treatment TD. The highest LRWC values under NI were observed in Salcedo INIA, which maintained its superiority in the other irrigation treatments; it should be noted there were no significant differences among genotypes in

DR or TD. However, Accession 50 had the lowest values in treatments NI and DR, and a highest value after Salcedo INIA and Accession 06 in irrigation treatment TD.

The highest values for specific leaf area (SLA) across the three irrigation treatments were for Accessions 26 and 50, but significant differences were not observed among genotypes for treatments NI and TD, while the average SLA value among treatments for the genotypes were similar. For stomatal density (SD) the average value for treatment NI was 32, but increased with increasing drought stress, reaching 34 in DR and 61 in TD. The highest NI values were observed for Accessions 20 and 43, while Accession 50 had the lowest values in the three experimental treatments.

### 3.3. Detection of Stress via Remote Sensing

Table 5 shows the Pearson correlation value r for grain yield with vegetation index (VI). A maximum correlation was observed from 79 to 89 DAS for the distinct vegetation indices with grain yield. The highest and directly proportional correlations for canopy coverage CC (r = 0.865) was at 80 DAS; for VI: NDRE (r = 0.85); TVI, CTVI, LCI, CIREDG, and CCCI (r = 0.84) was at 84 DAS and for the VI: NDVI, MRVI, IPVI, and NGRDI (r = 0.84); SAVI_05 and AVI (r = 0.83); RVI (r = 0.82); SA-VI_CC and GRNDVI (r= 0.81); GBNDVI (r = 0.80); NDVIG (r = 0.76); CIC and GRVI (r = 0.70) was 86 DAS. For the vegetation indices CREDG, RGR, GRE, and HI the correlations were negative. The vegetation indices CREDG (r = −0.85) and RGR (r = −0.84) had the highest negative correlation with yield. Similarly, in the correlation matrix containing spectroradiometric data there was a high correlation between grain yield and vegetation indices at 86 DAS (Table 6).

**Table 5.** Pearson r correlation matrix for grain yield with 24 vegetation indices and canopy coverage evaluated with UAV Panthom 4 multispectral for 18 days during vegetative development.

| VI\DAS | 52 | 61 | 63 | 65 | 66 | 70 | 74 | 77 | 79 | 80 | 82 | 84 | 86 | 89 | 93 | 98 | 115 | 129 |
|---|---|---|---|---|---|---|---|---|---|---|---|---|---|---|---|---|---|---|
| NDVI | 0.38 | 0.13 | −0.16 | 0.64 | 0.67 | 0.65 | 0.65 | 0.67 | 0.81 | 0.82 | 0.83 | 0.84 | 0.84 | 0.82 | 0.77 | 0.71 | 0.56 | 0.41 |
| TVI | 0.38 | 0.13 | −0.16 | 0.64 | 0.67 | 0.65 | 0.65 | 0.67 | 0.81 | 0.82 | 0.84 | 0.84 | 0.84 | 0.82 | 0.77 | 0.72 | 0.57 | 0.42 |
| RVI | 0.39 | 0.11 | −0.18 | 0.68 | 0.71 | 0.68 | 0.67 | 0.68 | 0.76 | 0.77 | 0.78 | 0.81 | 0.82 | 0.80 | 0.73 | 0.65 | 0.43 | 0.31 |
| MRVI | 0.38 | 0.13 | −0.16 | 0.64 | 0.67 | 0.65 | 0.65 | 0.67 | 0.81 | 0.82 | 0.83 | 0.84 | 0.84 | 0.82 | 0.77 | 0.71 | 0.56 | 0.41 |
| SAVI_CC | −0.44 | −0.45 | −0.36 | 0.51 | 0.59 | 0.62 | 0.61 | 0.64 | 0.73 | 0.72 | 0.74 | 0.76 | 0.80 | 0.78 | 0.72 | 0.63 | −0.62 | −0.84 |
| SAVI_05 | −0.27 | −0.32 | −0.26 | 0.56 | 0.62 | 0.64 | 0.63 | 0.65 | 0.77 | 0.78 | 0.80 | 0.82 | 0.83 | 0.81 | 0.78 | 0.72 | 0.56 | 0.39 |
| AVI | −0.33 | −0.35 | −0.26 | 0.54 | 0.61 | 0.64 | 0.62 | 0.64 | 0.76 | 0.76 | 0.79 | 0.81 | 0.82 | 0.80 | 0.78 | 0.72 | 0.55 | 0.36 |
| IPVI | 0.38 | 0.13 | −0.16 | 0.64 | 0.67 | 0.65 | 0.65 | 0.67 | 0.81 | 0.82 | 0.83 | 0.84 | 0.84 | 0.82 | 0.77 | 0.71 | 0.56 | 0.41 |
| CTVI | 0.38 | 0.13 | −0.16 | 0.64 | 0.67 | 0.65 | 0.65 | 0.67 | 0.81 | 0.82 | 0.84 | 0.84 | 0.84 | 0.82 | 0.77 | 0.72 | 0.57 | 0.42 |
| GNDVI | 0.12 | 0.02 | −0.13 | 0.36 | 0.46 | 0.57 | 0.59 | 0.59 | 0.69 | 0.71 | 0.74 | 0.75 | 0.76 | 0.73 | 0.75 | 0.60 | 0.44 | 0.32 |
| LCI | −0.34 | 0.24 | 0.08 | 0.44 | 0.56 | 0.64 | 0.64 | 0.63 | 0.70 | 0.72 | 0.79 | 0.84 | 0.84 | 0.82 | 0.77 | 0.71 | 0.56 | 0.53 |
| CIG | 0.11 | 0.00 | −0.15 | 0.36 | 0.46 | 0.58 | 0.59 | 0.59 | 0.65 | 0.66 | 0.68 | 0.68 | 0.70 | 0.66 | 0.69 | 0.52 | 0.35 | 0.25 |
| CIREDG | −0.38 | 0.25 | 0.12 | 0.30 | 0.46 | 0.63 | 0.64 | 0.61 | 0.68 | 0.70 | 0.77 | 0.84 | 0.83 | 0.81 | 0.77 | 0.71 | 0.55 | 0.54 |
| CCCI | −0.45 | 0.22 | 0.20 | −0.28 | −0.24 | 0.44 | 0.52 | 0.42 | 0.46 | 0.55 | 0.71 | 0.84 | 0.81 | 0.78 | 0.76 | 0.72 | 0.58 | 0.59 |
| NDRE | −0.38 | 0.24 | 0.12 | 0.30 | 0.46 | 0.62 | 0.63 | 0.61 | 0.68 | 0.71 | 0.78 | 0.85 | 0.84 | 0.82 | 0.77 | 0.72 | 0.57 | 0.56 |
| CREDG | 0.37 | −0.24 | −0.12 | −0.30 | −0.46 | −0.62 | −0.63 | −0.61 | −0.68 | −0.72 | −0.79 | −0.85 | −0.84 | −0.82 | −0.78 | −0.73 | −0.59 | −0.57 |
| CVI | 0.01 | −0.05 | −0.09 | −0.14 | −0.11 | 0.19 | 0.24 | 0.23 | 0.24 | 0.27 | 0.28 | 0.28 | 0.30 | 0.25 | 0.48 | 0.22 | 0.23 | 0.20 |
| GRVI | 0.11 | 0.00 | −0.15 | 0.36 | 0.46 | 0.58 | 0.59 | 0.59 | 0.65 | 0.66 | 0.68 | 0.68 | 0.70 | 0.66 | 0.69 | 0.52 | 0.35 | 0.25 |
| NGRDI | 0.08 | 0.08 | −0.01 | 0.50 | 0.56 | 0.60 | 0.64 | 0.67 | 0.82 | 0.81 | 0.81 | 0.82 | 0.84 | 0.83 | 0.61 | 0.71 | 0.39 | 0.02 |
| RGR | −0.09 | −0.08 | 0.00 | −0.48 | −0.54 | −0.59 | −0.63 | −0.67 | −0.82 | −0.81 | −0.81 | −0.82 | −0.84 | −0.82 | −0.63 | −0.72 | −0.39 | 0.02 |
| GRE | −0.30 | −0.29 | −0.10 | 0.13 | 0.32 | 0.29 | −0.33 | −0.28 | −0.50 | −0.53 | −0.62 | −0.70 | −0.68 | −0.67 | −0.66 | −0.43 | −0.49 | −0.46 |
| GBNDVI | 0.25 | 0.03 | −0.19 | 0.48 | 0.56 | 0.61 | 0.62 | 0.62 | 0.75 | 0.77 | 0.79 | 0.79 | 0.80 | 0.77 | 0.76 | 0.65 | 0.50 | 0.37 |
| GRNDVI | 0.19 | 0.06 | −0.16 | 0.55 | 0.61 | 0.63 | 0.64 | 0.64 | 0.76 | 0.77 | 0.79 | 0.80 | 0.81 | 0.79 | 0.76 | 0.66 | 0.49 | 0.36 |
| HI | −0.20 | 0.07 | 0.17 | −0.33 | −0.42 | −0.54 | −0.55 | −0.56 | −0.66 | −0.67 | −0.69 | −0.59 | −0.62 | −0.59 | −0.69 | −0.47 | −0.32 | −0.20 |
| CC | 0.31 | 0.36 | 0.26 | 0.52 | 0.53 | 0.72 | 0.66 | 0.65 | 0.85 | 0.87 | 0.86 | 0.84 | 0.83 | 0.82 | 0.80 | 0.79 | 0.77 | 0.73 |

The objective was to place the correlation coefficient (r) values close to 1 in green more intense, and close to −1 in red more intense.

**Table 6.** r-Pearson correlation coefficients between grain yield and 16 vegetation indices evaluated via spectroradiometry at three dates during quinoa crop development.

| DAS\IV | PRI | NDVI | TVI | RVI | MRVI | SAVI05 | IPVI | CTVI | GNDVI | LCI | CIG | CIREDG | NDRE | CVI | GBNDVI | GRNDVI |
|---|---|---|---|---|---|---|---|---|---|---|---|---|---|---|---|---|
| 73 | 0.36 | 0.57 | 0.56 | 0.65 | 0.59 | 0.59 | 0.57 | 0.56 | 0.4 | 0.44 | 0.35 | 0.53 | 0.43 | 0.03 | 0.44 | 0.52 |
| 86 | 0.82 | 0.8 | 0.8 | 0.82 | 0.8 | 0.8 | 0.8 | 0.8 | 0.65 | 0.8 | 0.6 | 0.78 | 0.8 | 0.28 | 0.73 | 0.76 |
| 100 | 0.65 | 0.69 | 0.69 | 0.64 | 0.7 | 0.7 | 0.69 | 0.69 | 0.52 | 0.65 | 0.41 | 0.56 | 0.61 | 0.23 | 0.6 | 0.63 |

The objective was to place the correlation coefficient (r) values close to 1 in green more intense, and close to −1 in red more intense.

Figure 7 provides vegetation indices (VI) calculated from data obtained by spectroradiometry at three dates: 73, 86, and 100 DAS. At 10 days post-stress (73 DAS) in DR and TD, the VI were the following: PRI, NDVI, TVI, RVI, MRVI, SAVI05, CTVI, GNDVI, LCI, CIREDG, NDRE, IPVI, CIG, CVI, GBNDVI, and GRNDVI and they significantly differentiated in the experimental field for irrigation treatment NI relative to DR and TD. In addition, at 23 days post-stress in TD and 7 in DR, the VI detected significant differences for the three irrigation treatments. At 100 DAS in DR and TD we observed no significant differences other than for CIG and CVI, which did significantly discriminate among the three irrigation treatments.

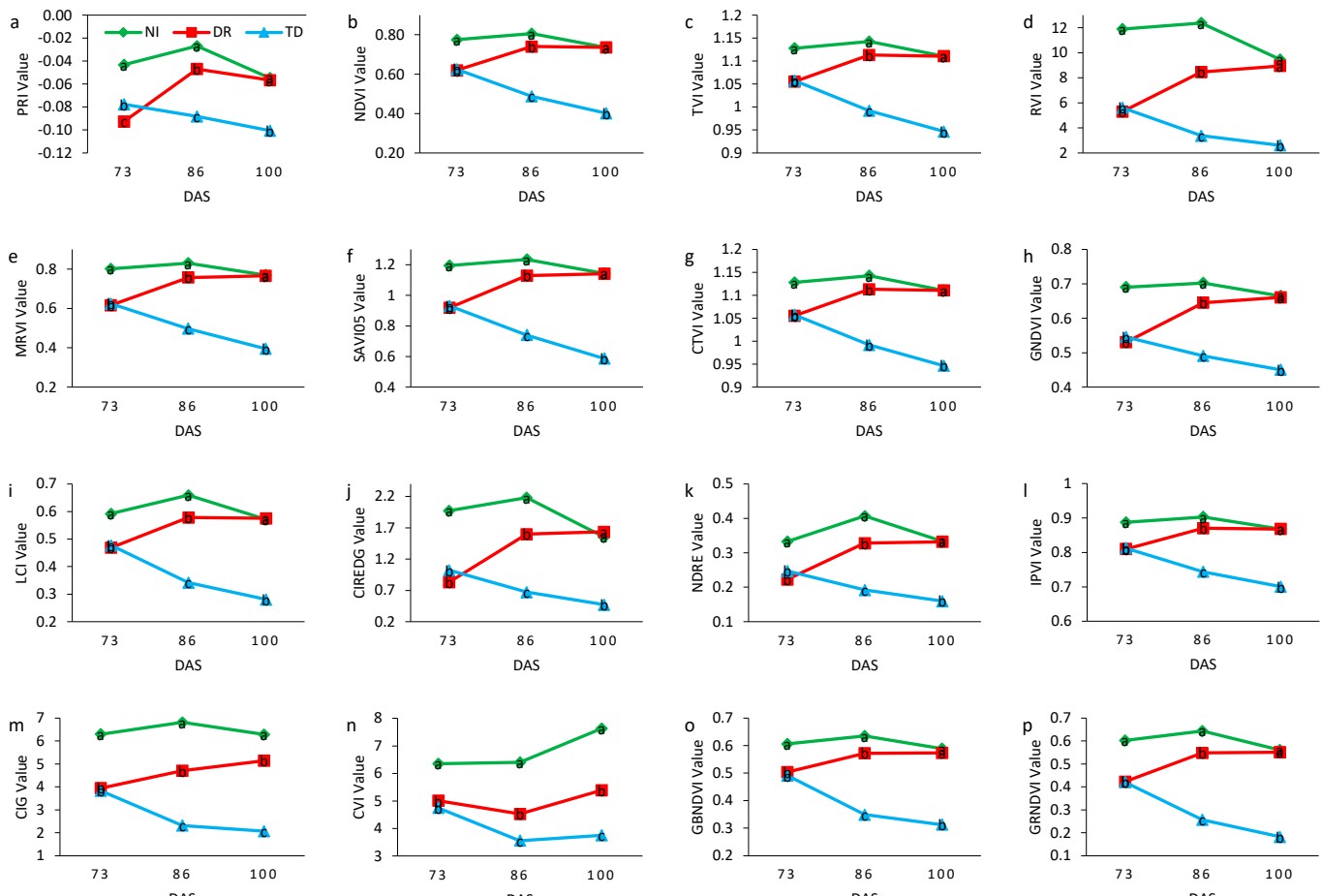

**Figure 7.** Changes in VI among treatments: (**a**) PRI, (**b**) NDVI, (**c**) TVI, (**d**) RVI, (**e**) MRVI, (**f**) SAVI05, (**g**) CTVI, (**h**) GNDVI, (**i**) LCI, (**j**) CIREDG, (**k**) NDRE, (**l**) IPVI, (**m**) CIG, (**n**) CVI, (**o**) GBNDVI and (**p**) GRNDVI at three time points during quinoa vegetative development as measured by spectroradiometry. Different letters signify significant differences among irrigation treatments ($p \leq 0.05$, Duncan's test).

### 3.4. Effect of Drought on Agronomic Variables

The agronomic variables grain yield, dry matter, harvest index, and water-use efficiency are presented in Table 7. Average yield of the genotypes in treatment NI was 3.8 t ha$^{-1}$ and we observed 33.16% and 93.68% reductions in the DR and TD treatments, respectively. The highest grain yield was attained by Accessions 42, 26, 49, Salcedo INIA, and Accession 43 in treatment NI. However, Accession 42 showed a 42.97% reduction in yield in treatment DR. In contrast, Salcedo INIA and Accession 26 had more modest yield reductions of 21.85% and 22.13%, respectively, in treatment DR. The lowest reduction in treatment TD was seen for Accession 50 (83.41%) and Salcedo INIA (90.27%), and these two lines had the highest grain yield in these highly drought-stressed conditions.

**Table 7.** Agronomic variables of the 10 genotypes under the three water-stressed treatments.

| Genotypes | Yield (t ha$^{-1}$) | | | DM (t ha$^{-1}$) | | | HI | | | WUE (kg m$^{-3}$) | | |
|---|---|---|---|---|---|---|---|---|---|---|---|---|
| | NI | DR | TD | NI | DR | DT | NI | DR | TD | NI | DR | TD |
| Accession 06 | 1.82 [d] | 1.01 [d] | 0.05 [d] | 11.76 [e] | 8.96 [de] | 3.38 [c] | 0.16 [e] | 0.11 [d] | 0.02 [d] | 1.56 [d] | 1.45 [d] | 1.78 [c] |
| Accession 20 | 3.08 [c] | 2.06 [c] | 0.10 [cd] | 13.27 [cd] | 10.69 [bcd] | 3.73 [bc] | 0.23 [d] | 0.19 [c] | 0.02 [d] | 1.76 [cd] | 1.74 [bcd] | 1.96 [bc] |
| Accession 23 | 4.10 [b] | 2.97 [abc] | 0.20 [cd] | 13.86 [bcd] | 11.57 [bcd] | 4.46 [ab] | 0.30 [c] | 0.26 [b] | 0.03 [cd] | 1.84 [bcd] | 1.88 [bcd] | 2.35 [ab] |
| Accession 26 | 4.97 [ab] | 3.87 [a] | 0.20 [cd] | 15.41 [abc] | 12.73 [abc] | 3.82 [bc] | 0.32 [bc] | 0.30 [a] | 0.05 [bcd] | 2.04 [abc] | 2.07 [abc] | 2.01 [bc] |
| Accession 42 | 5.26 [a] | 3.00 [abc] | 0.20 [cd] | 16.30 [ab] | 11.86 [abcd] | 4.28 [ab] | 0.32 [bc] | 0.25 [b] | 0.05 [bcd] | 2.16 [ab] | 1.93 [abcd] | 2.25 [ab] |
| Accession 43 | 4.28 [ab] | 2.89 [abc] | 0.10 [cd] | 17.86 [a] | 15.11 [a] | 4.57 [ab] | 0.24 [d] | 0.19 [c] | 0.02 [d] | 2.37 [a] | 2.45 [a] | 2.41 [ab] |
| Accession 49 | 4.95 [ab] | 3.00 [abc] | 0.30 [bc] | 13.59 [cd] | 9.14 [de] | 4.39 [ab] | 0.36 [ab] | 0.33 [a] | 0.07 [bc] | 1.80 [cd] | 1.48 [cd] | 2.31 [ab] |
| Accession 50 | 4.10 [b] | 2.39 [bc] | 0.68 [a] | 10.04 [e] | 7.04 [e] | 4.83 [a] | 0.41 [a] | 0.34 [a] | 0.14 [a] | 1.72 [cd] | 1.59 [cd] | 2.54 [a] |
| Blanca de Juli | 0.96 [d] | 0.67 [d] | 0.08 [cd] | 12.02 [de] | 9.69 [cde] | 3.81 [bc] | 0.08 [f] | 0.07 [d] | 0.02 [d] | 1.59 [d] | 1.57 [cd] | 2.00 [bc] |
| Salcedo INIA | 4.53 [ab] | 3.54 [ab] | 0.44 [b] | 15.73 [abc] | 13.47 [ab] | 4.83 [a] | 0.29 [cd] | 0.26 [b] | 0.09 [b] | 2.08 [abc] | 2.19 [ab] | 2.54 [a] |
| Mean | 3.80 | 2.54 | 0.24 | 13.98 | 11.03 | 4.21 | 0.27 | 0.23 | 0.05 | 1.89 | 1.84 | 2.22 |

In each column, common lowercase letters indicate groups of genotypes that were not significantly different ($p \leq 0.05$, Duncan's test).

The highest dry matter (DM) values were observed for Accessions 43, 42, Salcedo INIA, and Accession 26 both in the NI and DR treatments (Table 7). These also had the highest values along with Accessions 50, 49, and 23 in treatment TD, except for Accession 26, which showed the highest dry matter reduction at 75.21% under drought stress. On the other hand, Accession 50 had the lowest reduction at 51.89%. The average dry matter value for treatment NI was 13.98 t ha$^{-1}$ and was reduced by 21.10% and 69.86% in treatments DR and TD, respectively.

For harvest index (HI), the highest values in treatment NI were for Accessions 50 and 49, which maintained their superiority in treatment DR (Table 7). However, only Accession 50 presented a higher value in treatment TD and it was significantly different from the other genotypes.

In terms of water use efficiency (WUE), the average value under regime NI was 1.89 kg m$^{-3}$ and it increased in TD to 2.22 kg m$^{-3}$ (Table 7). Accessions 43, 42, 26, and Salcedo INIA had the highest values in treatment NI and retained their superiority in treatment DR. In TD, the highest WUE values were observed for Salcedo INIA and Accession 50.

### 3.5. Pearson r Correlation and Multivariate Analysis

Figure 8 shows the correlations among variables. We observed high positive correlations for yield (YLD) with HI (r = 0.92) and with BM (r = 0.88), while negative YLD correlations were measured with SD (r = −0.74) and with SPAD (r = −0.61). On the other hand, there were high positive correlations between BM and RWC (r = 0.91), PL and RWC (r = 0.89), TSW and DS (r = 0.89), and LA and HT (r = 0.87).

The results of the PCA are presented in Figure 9, in which the first two principal component axes explained 60.23% of the variance. The variables YLD, BM, HI, PL, SD, and RWC contributed more heavily to PC 1, while variables TSW, SPAD, HT, SLA, LA, and PD contributed more to PC 2 (Figure 9a). Additionally, Figure 9b depicts clear discrimination among the irrigation treatments, for example, the variables SPAD, SD, and WUE were more

highly associated with terminal drought while the variables YLD, BM, HI, PL, and HT were more highly expressed in normal irrigation and drought with recovery irrigation.

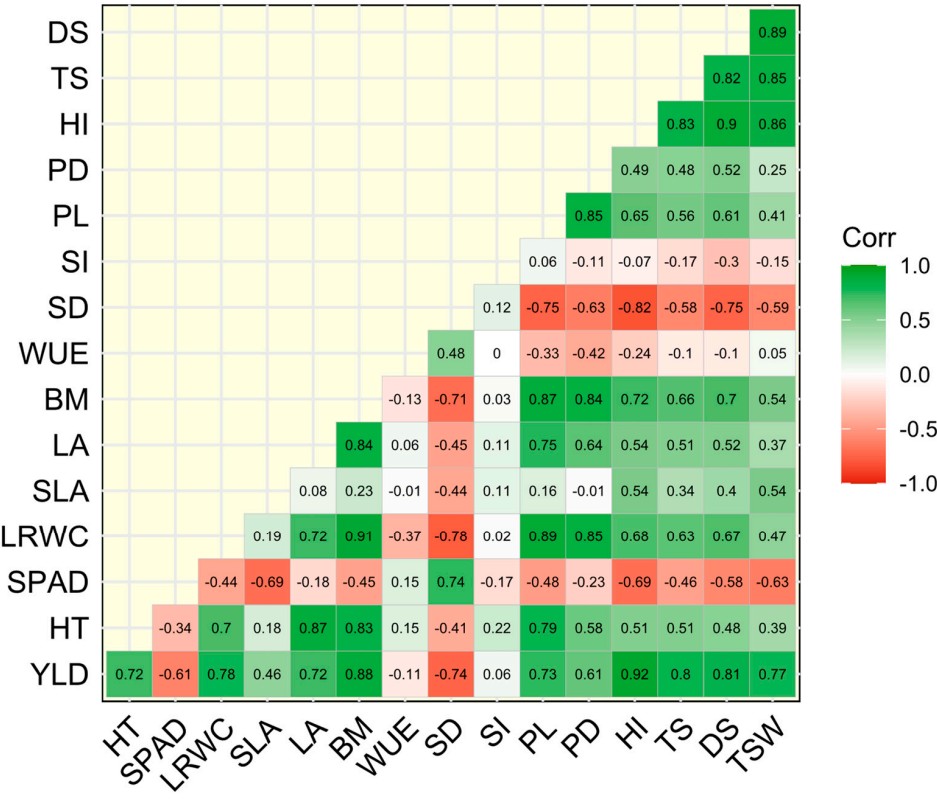

**Figure 8.** Pearson correlation r among morphological, physiological, and agronomic variables for 10 quinoa genotypes under three irrigation treatments. Abbreviations: DS = seed diameter; TS = seed thickness; HI = harvest index; PD = panicle diameter; PL = panicle length; SI = stomatal index; SD = stomatal density; WUE = water use efficiency; BM = biomass (dry); LA = leaf area; SLA = specific leaf area; LRWC = leaf relative water content; SPAD = chlorophyll; HT = plant height; YLD = seed yield; TSW = thousand seed weight.

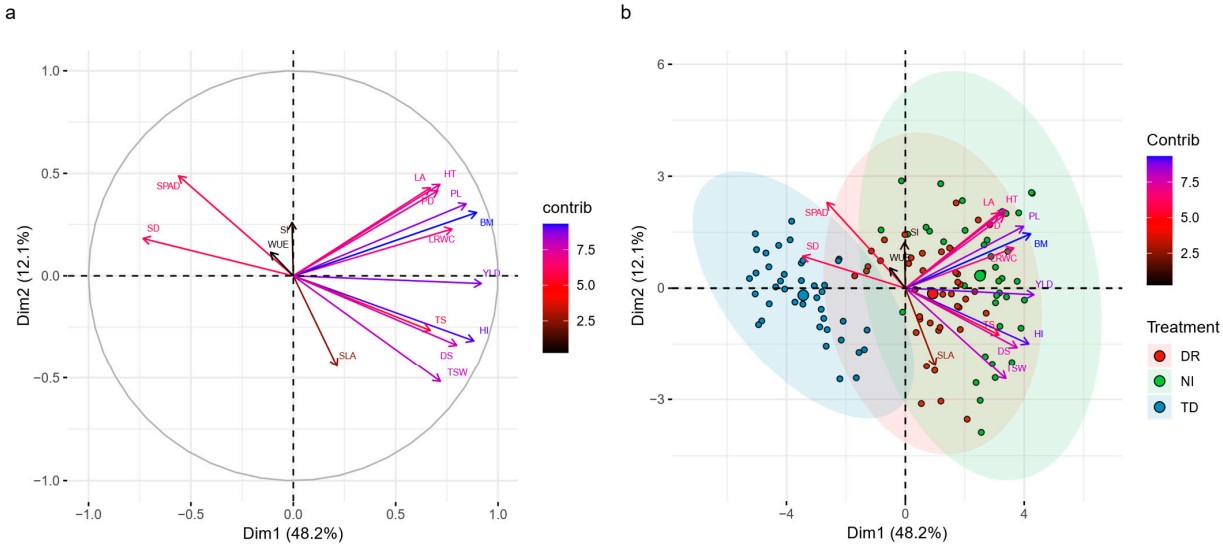

**Figure 9.** Principal component analyses (PCA) of the measured variables in 10 quinoa genotypes under three irrigation treatments. (**a**) PCA for the variables; (**b**) PCA for the irrigation treatments.

### 3.6. Selection Indices

In Table 8, the drought tolerance index (DTI) in treatment TD identified the superiority of the following genotypes: Accession 50, Salcedo INIA, and Accession 49. In treatment DR, the best-performing genotypes were Accessions 26, 42, Salcedo INIA, and 49. In contrast, for the drought susceptibility index (DSI) in treatment TD, Accession 50 had the lowest value and was significantly different from the rest of the lines, while in treatment DR there were no significant differences among the lines (Table 8).

**Table 8.** Drought tolerance index (DTI) and drought susceptibility index (DSI) calculated for the 10 genotypes under irrigation treatments DR and TD.

| Genotypes | DTI | | DSI | |
|---|---|---|---|---|
| | TD | DR | TD | DR |
| Accession 06 | 0.01 [d] | 0.12 [ef] | 1.03 [a] | 1.06 [a] |
| Accession 20 | 0.02 [d] | 0.44 [de] | 1.03 [a] | 0.98 [a] |
| Accession 23 | 0.05 [cd] | 0.84 [bc] | 1.01 [ab] | 0.79 [a] |
| Accession 26 | 0.07 [cd] | 1.33 [a] | 1.02 [ab] | 0.66 [a] |
| Accession 42 | 0.07 [cd] | 1.11 [ab] | 1.03 [ab] | 1.31 [a] |
| Accession 43 | 0.03 [d] | 0.88 [bc] | 1.04 [a] | 1.01 [a] |
| Accession 49 | 0.11 [bc] | 1.01 [abc] | 1.00 [ab] | 1.05 [a] |
| Accession 50 | 0.19 [a] | 0.67 [cd] | 0.88 [c] | 1.21 [a] |
| Blanca de Juli | 0.01 [d] | 0.04 [f] | 0.98 [ab] | 0.85 [a] |
| Salcedo INIA | 0.14 [ab] | 1.10 [ab] | 0.96 [b] | 0.62 [a] |
| Mean | 0.07 | 0.75 | 1.00 | 0.95 |

In each column, lowercase letters indicate significant differences among genotypes ($p \leq 0.05$, Duncan's test).

## 4. Discussion

The results indicate that irrigation treatment affected morphological characters including growth, leaf area, panicle length, and diameter, with increasing impact as drought stress intensified. Similar results were reported in other studies, for example with decreasing plant growth and leaf area in response to drought stress [20,22,24]. For physiological variables, negative effects were observed for relative water content, with significant increases in chlorophyll and stomatal density, while non-significant differences were observed for specific leaf area among irrigation treatments; the changes in specific leaf area that were observed were more attributable to the timing of water stress rather than its intensity [65]. However, changes in chlorophyll content depended both on the length of the stress period and the recuperation period [15]. Puangbut et al. [66] reported an increase in the SPAD index under water stress because chlorophyll density increased under drought conditions. The increase in chlorophyll density is due to the thicker leaves of plants being exposed to drought stress [67]. In this study, this is observed in the correlation of SPAD versus SLA, which is moderate and inverse (r = −0.69), i.e., the higher the SPAD, the lower the SLA, indicating greater leaf thickness (Figure 8). The increase in stomatal density in response to drought stress was also recorded by Issa Ali et al. [24] in quinoa cultivar 'L119', while in the cultivar 'Titicaca' did not show significant differences between drought and control treatments. Among genotypes, we observed the lowest stomatal density in Accession 50 and this did not change considerably among the three treatment regimes, an apparently intrinsic characteristic of this genotype. According to Zhang et al. [25] lower stomatal density should translate into less water loss and is therefore another characteristic of drought-tolerant cultivars. In addition to plant height and leaf area variables, other traits such as leaf area index, leaf tilt, and leaf stem angle can significantly reflect drought [68]. Therefore, it is important to consider these variables in future research.

We noted a high positive correlation between vegetation indices and grain yield at 79–89 DAS, with the highest correlation being observed for canopy coverage CC (r = 0.865) at 80 DAS; however, for vegetation indices in general, the highest correlation was observed

from 84–86 DAS. Similarly, the correlations calculated from remote spectroradiometric data were highest at 86 DAS for all vegetation indices. This period coincides with the phenological stage of anthesis in quinoa grown in this Peruvian coastal region. In this same period, the chlorophyll content and canopy coverage reached their highest levels, which likely influenced the high correlation coefficients. Therefore, the optimum time for conducting remote sensing research using unmanned aerial vehicles (UAV) and spectroradiometric equipment to produce predictive quinoa yield models would be from 79–89 DAS. The vegetation indices combining NIR/visible or NIR/near-red had the best correlations with yield [69] therefore, NDVI should be the most useful for predicting quinoa grain yield [70].

In the same vein, the vegetation indices PRI, NDVI, TVI, RVI, MRVI, SAVI05, CTVI, GNDVI, LCI, CIREDG, NDRE, IPVI, CIG, CVI, GBNDVI, and GRNDVI were all sufficiently sensitive to detect the effects of drought (Figure 7). The PRI index discriminated between healthy and stressed plants and could therefore be used as a robust index for evaluating moderate water stress [71]. Ihuoma y Madramootoo [72] mentioned indices PRI and OSAVI as being the most sensitive for identifying levels of water stress in tomato plants; similarly, Sankaran et al. [29] identified NDVI and GNDVI as indices that can differentiate between irrigated and non-irrigated plots. The CIG also clearly identified the portion of the field that had not been irrigated [27]. In general, vegetation indices obtained from remote sensors like UAV and spectroradiometry provide dependable data and may be adopted as rapid and affordable approaches to evaluate stress and phenotype plants [29,71,73–75].

In terms of grain and dry matter yield, there is a significant decrease depending on the level of stress. Our results are similar to those obtained by Razzaghi et al. [22], who reported 0.26 t ha$^{-1}$ and 4.71 t ha$^{-1}$ of grain and dry matter yield, respectively, under water stress. Similarly, other studies have reported a decrease in dry matter [22,24,76]. Additionally, there is a high direct correlation between grain yield and dry matter, leaf relative water content, and grain attributes [9,19]. The grain yield of Accession 50 under terminal drought treatment was higher than that of the commercial genotypes Salcedo INIA and Blanca de Juli.

Based on drought tolerance (DTI) and susceptibility (DSI) indices, among quinoa genotypes there is considerable variation for drought tolerance [9]. We observed that not all genotypes are severely affected based on the intensity of stress; for genotypes like Accession 50, the variables we evaluated were not severely affected based on the level of stress. We observed lower chlorophyll values, stomatal density, and leaf area for this strain; in contrast, taller plants, greater specific leaf area, increased leaf relative water content, more dry matter, higher harvest index, and more efficient water use in comparison with other genotypes evaluated in terminal drought stress conditions that were severely affected. Characteristics such as reduced leaf area and stomatal density that helped reduce water loss from the leaves [77,78] possibly accounted for Accession 50's stable and high water content relative to the other genotypes, along with its higher water use efficiency. We assume that these traits resulted in higher dry matter content and grain yield under terminal drought stress and so this would be a promising genotype for production under water stress conditions.

## 5. Conclusions

Terminal drought and drought with recovery irrigation alter the morphology and physiology of quinoa plants. At the morphological level, leaf area, panicle length, and plant height were reduced by 62.7%, 58%, and 27.2%, respectively, under terminal drought conditions, while physiological changes including reductions in leaf relative water content and increases in chlorophyll content and stomatal density. For agronomic factors, grain yield and total dry biomass were reduced by 93.7% and 69.9% respectively, under terminal drought stress conditions. As such, selection indices DTI and DSI indicated that Accession 50, Salcedo INIA and Accession 49 responded well under conditions of terminal drought, while Accessions 26, 42, Salcedo INIA, and 49, were well adapted to drought stress with recovery irrigation.

Tools for real-time remote sensing, such as multispectral cameras and spectroradiometry, facilitate data acquisition for monitoring stress and phenotyping under stress conditions in a rapid and precise manner. In this sense, the 24 vegetation and canopy coverage indices calculated via multispectral imaging are highly correlated with grain yield between 79 and 89 DAS, such that this time period is optimal for using remote sensing to formulate predictive yield models for quinoa. On the other hand, the 16 vegetation indices calculated via spectroradiometry were informative for discriminating effects of irrigation treatments beginning at 10 days after initiation of drought stress.

This study evaluated the use of vegetation spectral indices to detect drought water stress in quinoa plants under terminal drought and recovery irrigation. Such induced water deficit negatively affected crop yield and other traits. Our study with multispectral cameras showed that NDRE (r = 0.85) and CREDG (r = 0.86) have the highest correlations of spectral indices with yield, so they may be the most sensitive indices to distinguish water stress levels in quinoa plants grown under different irrigation conditions. Likewise, measurements made with the spectroradiometer indicate that PRI (r = 0.82) and RVI (r = 0.82) are most highly correlated with yield.

**Author Contributions:** Conceptualization, M.E.M.-A., A.A.-A. and R.O.G.-R.; methodology, R.O.G.-R. and N.E.L.-C.; software, N.E.L.-C., F.C.L.-C., A.A.S.-J. and O.Z.; validation, N.E.L.-C., F.C.L.-C. and A.A.S.-J.; formal analysis, N.E.L.-C., F.C.L.-C., A.A.S.-J. and R.O.G.-R.; investigation, N.E.L.-C., F.C.L.-C., A.A.S.-J., R.O.G.-R. and M.E.M.-A.; resources, N.E.L.-C., F.C.L.-C., A.A.S.-J. and M.E.M.-A.; data curation, N.E.L.-C. and O.Z.; writing—original draft preparation, N.E.L.-C., F.C.L.-C. and A.A.S.-J.; writing—review and editing, N.E.L.-C., R.O.G.-R., E.N.J. and N.C.H.; visualization, N.E.L.-C. and R.O.G.-R.; supervision, M.E.M.-A., A.A.-A. and R.O.G.-R.; project administration, N.W.L.-C. and M.E.M.-A.; funding acquisition, M.E.M.-A. All authors have read and agreed to the published version of the manuscript.

**Funding:** This research was financed under the project "Fenotipaje de Alta Frecuencia en el Cultivo de Quinoa para la Caracterización de Germoplasma y el Mejoramiento Genético" by the Universidad Nacional de San Agustín de Arequipa with contract number IBA-IB-09-2021-UNSA.

**Data Availability Statement:** The original contributions presented in the study are included in the article, further inquiries can be directed to the corresponding author.

**Acknowledgments:** We thank to Henry Gustavo Polanco Cornejo, and to Héctor Medina Dávila, Facultad de Agronomía de la Universidad Nacional de San Agustín de Arequipa for access to field space for installing the experiment within the Centro de Investigación, Enseñanza y Producción Agrícola. We are likewise grateful to Hildo Loaiza Loza for his support in processing the multispectral images.

**Conflicts of Interest:** The authors declare no conflicts of interest.

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
