# Peer review of "Phenotyping for Effects of Drought Levels in Quinoa Using Remote Sensing Tools"

_agronomy, doi:10.3390/agronomy14091938_

Round 1

Reviewer 1 Report

Comments and Suggestions for Authors

The article demonstrates that remote sensing tools are an effective method for phenotyping quinoa genotypes under drought conditions. The authors measured various morphological and physiological traits in 10 different quinoa genotypes across three water regimes (normal irrigation, drought and recovery, and terminal drought) and vegetation indices using a multispectral remote sensor to identify more drought-tolerant genotypes. The study is well-executed; however, my primary concern is the lack of precipitation data, which is crucial for a drought experiment. Additionally, the experiment was conducted over only one growing season, which may not account for year-to-year environmental variations. A minimum of two years of experimentation is necessary to draw stronger conclusions. Despite this, the authors effectively measured various traits.

Furthermore, I have highlighted a few comments and suggestions that could improve the manuscript.

Methods

Line 79: There is no data about precipitation. Since this is a drought experiment, it would be beneficial to include this important parameter, ideally presented in a graph covering the growing season.

Line 105: The fertilization was done from 15 DAS until 63 DAS. How many times was fertilization applied? Was it done daily with irrigation?

Line 110: Was irrigation done every day? Were there any days without rain? Including precipitation data is important.

Line 112: On which day after sowing was the experiment harvested? Did some genotypes mature at different times? Were all genotypes harvested simultaneously? If not, different amounts of water might have been given to different genotypes under normal irrigation.

Line 177: It should be Table 3, not Table 2. Please correct this error.

Line 180: It would be helpful to include the model of the equipment used.

Results:

Figure 1: The data shown is the average from all 10 genotypes. As shown in Table 4, not all genotypes exhibit the same behavior regarding plant height. It might be beneficial to include standard error bars or deviation bars to illustrate the variability.

Line 251 and Line 254: What do "LP" and "DP" refer to? Perhaps they should be "PL" (Panicle Length) and "PD" (Panicle Diameter) as mentioned in line 119. Please correct this mistake in line 254 and in Table 4. I suggest adding the meanings of "PL" and "PD" in a note for Table 4 to clarify the abbreviations.

Line 283: What do you mean by "not shown"? Is it not present in Table 5? However, in line 290, the authors describe this trait.

Line 289: After "accession 06" and "Salcedo INIA."

Line 333: What does "CC" stand for? Canopy Coverage? Table 3 lists 25 indices, but Table 6 reports only 24. The PRI index is missing.

Line 307: There are two periods in the sentence. It might be worth mentioning that CREDG and RGR have the highest negative correlation with yield.

Line 315: Change "quinua" to "quinoa." In various parts of the document, the authors wrote "quinua," but it should be "quinoa" in English, as seen in figures, tables, and the entire discussion section.

Line 318: What do "RR" and "ST" stand for?

Line 363: It should be "SD" (Stomatal Density) instead of "DE." I'm not sure what "DE" refers to.

Discussion:

SPAD measurements being higher in drought treatments is illogical. You mentioned this could be to maintain a high photosynthesis rate in drought-tolerant genotypes (Line 407), but in Table 5, all genotypes, including the very susceptible ones, show increased values. The higher SPAD values in the drought treatment might be because the measurements were taken from young leaves at different levels. It's also worth analyzing at different time points since Table 5 reports the average of all measurements (44, 59, 70, 83, and 106 DAS).

Author Response

Comments 1: Line 79: There is no data about precipitation. Since this is a drought experiment, it would be beneficial to include this important parameter, ideally presented in a graph covering the growing season.

Response 1: Figure 1 includes daily precipitation data for the period July 1, 2023 to December 31, 2023. In that period, the accumulated precipitation was 3.1 mm. This data is mentioned in the document. It is worth mentioning that the research area is considered arid.

Comments 2: Line 105: The fertilization was done from 15 DAS until 63 DAS. How many times was fertilization applied? Was it done daily with irrigation?

Response 2: Fertilization was performed daily through irrigation from 15 DAS to 63 DAS. The manuscript was edited to clarify this point in the document.

Comments 3: Line 110: Was irrigation done every day? Were there any days without rain? Including precipitation data is important.

Response 3: Irrigation was performed daily according to the treatments. This is now specified in the revised document. In addition, the daily irrigation application is presented in Figure 3.

Comments 4: Line 112: On which day after sowing was the experiment harvested? Did some genotypes mature at different times? Were all genotypes harvested simultaneously? If not, different amounts of water might have been given to different genotypes under normal irrigation.

Response 4: The quinoa genotypes were harvested according to the physiological maturity reached; this information on the harvest of genotypes is now included in the revised document. It is worth mentioning that, to calculate the WUE, the date of harvest and the water applied up to that date were considered.

Comments 5: Line 177: It should be Table 3, not Table 2. Please correct this error.

Response 5: The error was corrected. A previous table (table 2) was deleted, so table 3 in the last document now appears as table 2.

Comments 6: Line 180: It would be helpful to include the model of the equipment used.

Response 6: The model of equipment used was included.

Comments 7: Figure 1: The data shown is the average from all 10 genotypes. As shown in Table 4, not all genotypes exhibit the same behavior regarding plant height. It might be beneficial to include standard error bars or deviation bars to illustrate the variability.

Response 7: The figure with the standard deviation is included in the revision.

Comments 8: Line 251 and Line 254: What do "LP" and "DP" refer to? Perhaps they should be "PL" (Panicle Length) and "PD" (Panicle Diameter) as mentioned in line 119. Please correct this mistake in line 254 and in Table 4. I suggest adding the meanings of "PL" and "PD" in a note for Table 4 to clarify the abbreviations.

Response 8: Replaced “PL” in place of “LP” and “PD” in place of “DP” in document. Also, the meanings of 'PL' and 'PD' were included in the note to table 3. With the deletion of table 2, table 4 is now table 3.

Comments 9: Line 283: What do you mean by "not shown"? Is it not present in Table 5? However, in line 290, the authors describe this trait.

Response 9: The phrase “not shown” was removed from the document.

Comments 10: Line 289: After "accession 06" and "Salcedo INIA."

Response 10: The description in that paragraph has been improved: “The value of Accession 50 is higher after Salcedo INIA and Accession 06 in TD for LRWC.”

Comments 11: Line 333: What does "CC" stand for? Canopy Coverage? Table 3 lists 25 indices, but Table 6 reports only 24. The PRI index is missing.

Response 11: The meaning of “CC” (canopy coverage) was placed in the document. Table 5 reports the 24 indices calculated from multispectral images. PRI was not calculated with multispectral images, because a specific wavelength is needed; therefore, PRI was only calculated with data obtained from the spectrometer, which is shown in Table 6. All this is clarified in the revised document.

Comments 12: Line 307: There are two periods in the sentence. It might be worth mentioning that CREDG and RGR have the highest negative correlation with yield.

Response12: The modification is made where it is mentioned that “CREDG and RGR have the highest negative correlation with performance”.

Comments 13: Line 315: Change "quinua" to "quinoa." In various parts of the document, the authors wrote "quinua," but it should be "quinoa" in English, as seen in figures, tables, and the entire discussion section.

Response 13: The error has been corrected.

Comments 14: Line 318: What do "RR" and "ST" stand for?

Response 14: The acronyms have been corrected, “RR” has been replaced by “DR”, and “ST” by “TD”. Thanks for catching this.

Comments 15: Line 363: It should be "SD" (Stomatal Density) instead of "DE." I'm not sure what "DE" refers to.

Response 15: The acronym was corrected, replacing “DE” by “SD” (stomatal density).

Comments 16: SPAD measurements being higher in drought treatments is illogical. You mentioned this could be to maintain a high photosynthesis rate in drought-tolerant genotypes (Line 407), but in Table 5, all genotypes, including the very susceptible ones, show increased values. The higher SPAD values in the drought treatment might be because the measurements were taken from young leaves at different levels. It's also worth analyzing at different time points since Table 5 reports the average of all measurements (44, 59, 70, 83, and 106 DAS).

Response 16: . The wordCing of the above paragraph has been improved, two new references have been included to support the increase of SPAD chlorophyll content in drought, they refer to increase in chlorophyll density due to the thicker leaves of plants exposed to drought stress.

Reviewer 2 Report

Comments and Suggestions for Authors

Drought research is important in production of agricultural crops. This manuscript studied the

morphological and physiological changes in ten quinoa genotypes under drought stress. The special comments are as follows:

1 the abstract and conclusion were lack of data result.

2 the authors calculated many traits for quinoa, well done!

3 the image of qiunoa plants acquired by UAV should be provided in section 2.1.

4 in section 2.4, why you choose plant and leaf area, the reason should be specified.

5 Multispectral images under different drought stress should be provided in section 2.6.

6 Agricultural information technology plays an important role in the study of crop phenotypes and genotypes. How to recognize drought using machine learning technology is a more significant work in plant breeding and field management. Thus, in introduction or discussion section, the research work in drought recognition for Agricultural information field should be introduced from the perspective of machine learning technology. The following references may helpful for authors.

(1) Calculation method of wilting index based on fractal dimension of multispectral images for the soybean canopy, Computers and Electronics in Agriculture, 2023

(2) A Fourier Transform-Based Calculation Method of Wilting Index for Soybean Canopy Using Multispectral Image, Agronomy, 2022

(3) Drought recognition based on feature extraction of multispectral images for the soybean canopy, Ecological Informatics, 2023

7 Besides plant height and leaf area in this study, other important traits can reflect drought significantly (leaf area index, leaf inclination and leaf stalk angle), which can be researched in the future work in discussion section.

(1) Dynamic simulation of leaf area index for the soybean canopy based on 3D reconstruction, Ecological Informatics, 2023

 This paper was well organized, thus, I suggest minor revision after addressing the above  comments.

Comments on the Quality of English Language

moderate editing

Author Response

Comments 1: The abstract and conclusion were lack of data result.

Response 1: The abstract and conclusion were improved, including addition of the most notable results data.

Comments 2: The authors calculated many traits for quinoa, well done!

Response 2: Thank you for your positive comments!

Comments 3: The image of quinoa plants acquired by UAV should be provided in section 2.1.

Response 3: Figure 2, titled "UAV image of the experimental field and plots 63 days after sowing", has been included in Section 2.3.

Comments 4: In section 2.4, why you choose plant and leaf area, the reason should be specified.

Response 4: Morphological variables, such as plant height and leaf area, are direct indicators of water stress. For this reason, these variables were measured in the research carried out.

Comments 5: Multispectral images under different drought stress should be provided in section 2.6.

Response 5: Figure 4, titled “Normalized Difference Vegetation Index (NDVI) image of the plots under three irrigation treatments: (a)  NDVI image at 63 DAS (1 day before the start of drought treatment); (b) NDVI image at 77 DAS (14 days after drought); (c) NDVI image at 86 DAS (23 days after drought for TD and 9 days after recovery irrigation for DR)”, has been included in Section 2.6.

Comments 6: Agricultural information technology plays an important role in the study of crop phenotypes and genotypes. How to recognize drought using machine learning technology is a more significant work in plant breeding and field management. Thus, in introduction or discussion section, the research work in drought recognition for Agricultural information field should be introduced from the perspective of machine learning technology. The following references may helpful for authors.

(1) Calculation method of wilting index based on fractal dimension of multispectral images for the soybean canopy, Computers and Electronics in Agriculture, 2023

(2) A Fourier Transform-Based Calculation Method of Wilting Index for Soybean Canopy Using Multispectral Image, Agronomy, 2022

(3) Drought recognition based on feature extraction of multispectral images for the soybean canopy, Ecological Informatics, 2023

Response 6: The discussion section has been improved and one of the recommended references has been included.

Comments 7: Besides plant height and leaf area in this study, other important traits can reflect drought significantly (leaf area index, leaf inclination and leaf stalk angle), which can be researched in the future work in discussion section.

(1) Dynamic simulation of leaf area index for the soybean canopy based on 3D reconstruction, Ecological Informatics, 2023

Response 7: The suggestion has been included into the discussion section, supported by the recommended reference: “Dynamic simulation of leaf area index for soybean canopy based on 3D reconstruction.”

Reviewer 3 Report

Comments and Suggestions for Authors

This manuscript used remote sensing for Quinoa phenotyping under drought stress. The manuscript was well-organized and introduce an interesting topic. Some suggestions are provided for your consideration.

1. Please provide some multi-spectral images of studied Quinoa samples, in the Materials and Methods Section.

2. Table 2 is not easy to understand. please add more details.

3. Eq. (1), please add a '%'.

4. Please kindly discuss about the limitation of this study and highlight the novelty of this study. And also discuss if the combination of the UAV and multispectral imaging can realized automatic phenotyping of Quinoa plants.

Comments on the Quality of English Language

None.

Author Response

Comments 1: Please provide some multi-spectral images of studied Quinoa samples, in the Materials and Methods Section.

Response 1: Figure 1 and Figure 4 were included in the Materials and Methods section.

Comments 2: Table 2 is not easy to understand. please add more details.

Response 2: Table 2 was removed and figure 3 “Daily application of water for the three irrigation treatments. Abbreviations: NI = normal irrigation (NI); DR = drought with recovery irrigation and TD = terminal drought“ was included.

Comments 3: Eq. (1), please add a '%'.

Response 3: It has been corrected as suggested.

Comments 4: Please kindly discuss about the limitation of this study and highlight the novelty of this study. And also discuss if the combination of the UAV and multispectral imaging can realized automatic phenotyping of Quinoa plants.

Response 4: The wording of the conclusions has been improved, highlighting the novelties of the study, as well as highlighting some aspects that may be important to understand the contribution of the study.

Round 2

Reviewer 1 Report

Comments and Suggestions for Authors

Thank you to the authors for considering my comments and suggestions. The manuscript is now acceptable for publication with the changes that have been made.